# Point Cloud Densification for 3D Gaussian Splatting from Sparse Input Views

## ABSTRACT

The technique of 3D Gaussian splatting (3DGS) has demonstrated its effectiveness and efficiency in rendering photo-realistic images for novel view synthesis. However, 3DGS requires a high density of camera coverage, and its performance inevitably degrades with sparse training views, which significantly restricts its applications in real-world products. In recent years, many researchers have tried to use depth information to alleviate this problem, but the performance of their methods is sensitive to the accuracy of depth estimation. To this end, we propose an efficient method to enhance the performance of 3DGS with sparse training views. Specifically, instead of applying depth maps for regularization, we propose a densification method that generates high-quality point clouds for improved initialization of 3D Gaussians. Furthermore, we propose Systematically Angle of View Sampling (SAOVS), which employs Spherical Linear Interpolation (SLERP) and linear interpolation for side view sampling, to determine unseen views outside the training data for semantic pseudo-label regularization. Experiments show that our proposed method significantly outperforms other promising 3D rendering models on the ScanNet dataset and the LLFF dataset. In particular, compared with the conventional 3DGS method, the PSNR and SSIM performance gains achieved by our method are up to 1.71dB and 0.07, respectively. In addition, the novel view synthesis obtained by our method demonstrates the highest visual quality with fewer distortions.

## CCS CONCEPTS

• **Computing methodologies → 3D imaging**.

## KEYWORDS

3D Gaussian Splatting, Sparse Input Views, Semantic Knowledge Prior

## 1 INTRODUCTION

Neural Radian Field (NeRF) [20] has achieved remarkable successes in rendering novel-view images, by estimating the volumetric density and color values of voxels with a multilayer perception (MLP) and rendering the voxels for synthesizing the corresponding novel-view images. This technique demonstrates high industrial values in real-world applications, including augmented reality (AR) and virtual reality (VR), and attracted researchers' attention in the past

*ACM MM, 2024, Melbourne, Australia*
© 2024 Copyright held by the owner/author(s). Publication rights licensed to ACM.
ACM ISBN 978-x-xxxx-xxxx-x/YY/MM
https://doi.org/10.1145/nnnnnnn.nnnnnnn

**Unpublished working draft. Not for distribution.**

years. To reconstruct high-quality 3D scenes, NeRF typically requires many images captured by cameras from different views. However, in many situations, it is hard to deploy such a dense camera coverage to capture images from the scenes. Therefore, many researchers [4, 9, 13, 25, 31, 38] attempt to reconstruct 3D scenes with images from sparse views, which is a significantly challenging problem.

In NeRF, input images from different views can be viewed as constraints in 3D scene reconstruction, which aims at solving the ambiguity issue of 3D content generation from its corresponding 2D projection images [14]. Previous studies [32, 35] show that when processing input images of sparse views, existing methods usually produce degraded 3D scene content, corrupted by undesirable floaters. To handle this issue, Roessle *et al.* [25] first applied a depth prediction method to obtain dense maps and add depth regularization to the training process to enhance the performance of NeRF. Similarly, DS-NeRF [9] adopts the structure from motion (SfM) method to obtain sparse depth information, which is used to supervise the NeRF optimization process for reconstruction. These two works demonstrate that incorporating depth information can effectively improve the performance of NeRF in sparse-input-view settings. However, the performance of such depth-guidance methods is sensitive to the accuracy of the depth estimation. In other words, inaccurate depth maps used in these methods inevitably produce artifacts in the synthesized novel view images. Alternatively, DietNeRF [13] utilizes a pretrained CLIP-ViT [22] to encode the target objects and images captured from unobservable views into the latent space and introduces a semantic consistency loss to regularize these latent features. SinNeRF [36] proposes to reconstruct 3D scenes from a single image by adding semantic regularization in the training loss. Specifically, this method constructs pseudo labels for side views, which are randomly generated next to the training view, by using semantic prior knowledge and a local texture discriminator. However, DietNeRF and SinNeRF cannot provide a feasible scheme for side-view sampling for real-world scenes, i.e. images in the ScanNet and LLFF datasets.

Recently, 3D Gaussian Splatting (3DGS) [14], which models 3D scenes with a set of 3D Gaussians, has shown its effectiveness in accelerating scene optimization and real-time 3D rendering. Chung *et al.* [7] introduced a depth rendering method for 3DGS, which applies $\alpha$-blending on the projected depth value and then regularizes the depth with the $L_1$ distance calculated by the pretrained depth estimation network. However, estimated depth maps and rendered depth maps may have different scales, introducing noise to the optimization process if $L_1$ distance is used for depth regularization. [41]. Instead of $L_1$ loss, the studies [35, 41] use the Pearson correlation as the measurement for depth regularization. These methods rely on powerful knowledge prior to generate depth maps for depth regularization while failing to acquire accurate depth values and

                                    

imposing noise to the optimization process if depth maps are not accurate.

In this paper, we propose a robust method for synthesizing images of novel views by optimizing 3D scenes with an enhanced loss function, regularized by additional semantic terms encoded from side views of the training views, and effectively incorporating depth prior to the 3DGS method. In our method, we employ a joint learning method to optimize the 3DGS model with the supervision of the training views and the side views The previous studies, DietNeRF [13] and SinNeRF [36], show the effectiveness of adopting unseen view information in 3D view optimization. However, these methods do not provide a generic side-view sampling scheme for real-world scenes, i.e. images in the ScanNet and LLFF datasets. To utilize the benefit of these unseen views, we propose the Systematically Angle of View Sampling (SAOVS) method to adaptively select side views of the training views. Specifically, we adopt spherical linear interpolation (SLERP) [30] to parameterize side-view directions and linear interpolation to parameterize the position of the cameras. Side views, sampled with SAOVS, cover a large portion of the 3D scene while having very similar views to the training views. The optimization process is guided by training images and regularizing latent features extracted from training views and side views. The second challenge is that conventional methods, adding a depth regularization term in the training loss, introduce noise to the optimization process if the depth information is erroneous. Meanwhile, we observed that the performance of 3DGS is sensitive to the initialization status. As revealed by the previous study [41], given the same number of training images, the performance of 3DGS degrades if the number of SfM points decreases. To handle these issues, we propose a point cloud densification method applied before 3D Gaussian initialization that effectively incorporates depth prior with 3DGS and alleviates the negative impact due to the incorrect depth values. In practice, we apply a pretrained Dense Prediction Transformer (DPT) [1, 23, 24] to obtain a dense map and then lift a portion of the depth values to the 3D world space with synthetic rays, resulting in a high-quality point cloud. The high-quality point cloud is used for the initialization of 3DGS.

The contributions of our work can be summarized as follows:

(1) We propose a point cloud densification method to improve the quality of point clouds that are used for initializing 3D Gaussians.
(2) We propose Systematically Angle of View Sampling (SAOVS) to sample side views for semantic pseudo-label boosted training.
(3) With the point cloud densification method and SAOVS, depth prior information can be effectively and efficiently aggregated with 3DGS, allowing our method to outperform the previous methods on the ScanNet [8] dataset and the LLFF [18] dataset.

## 2 RELATED WORKS

### 2.1 NeRF From Sparse Input Views

NeRF from sparse input views is a challenging problem as the inputs provide less information to supervise the optimization process. Most of the methods for this task rely on constructing regularization terms for training loss by using some knowledge priors, e.g. semantic prior of the scene and depth prior, acquired with SfM method or other depth estimation methods. PixelNeRF [38] feeds the deep features of query points, extracted by a pre-trained convolutional neural network, to the NeRF network, facilitating the prediction of density and color. DietNeRF [13] introduces an auxiliary semantic consistency loss, measuring the distance of embedding encoded by a pre-trained CLIP-ViT [22] and encouraging the recovery of scene geometry. SinNeRF [36] not only utilizes a similar semantic consistency loss for reconstructing 3D models with better global structure but also adopts an adversarial loss to restore the local texture of the scene. To exploit depth information on sparse-input-view problems, RegNeRF [21] introduces a geometry regularization to encourage depth smoothness from unseen views. DS-NeRF [9] directly supervises NeRF optimization with sparse depth information obtained from point clouds. Concurrently, Roessle *et al.* [25] applied a pre-trained dense completion network to construct dense depth maps from sparse depth information and regularize the training loss with the distance between estimated depth maps and rendered depth maps. Recently, the state-of-the-art method ViP-NeRF [31], built upon DS-NeRF, used additional visibility prior as relative depth information. Compared with dense depth maps, visibility relaxes the constraints on absolute depths, which is helpful when the depth is incorrect, and is beneficial for reconstructing NeRF models with good quality.

### 2.2 3D Gaussian Splatting Methods

The emergence of 3DGS not only marks the performance improvement of novel-view rendering techniques but also represents a successful attempt to search for new 3D scene representations. Various applications based on 3DGS appeared soon after the announcement of 3DGS. Recent methods [6, 15, 37] have aggregated diffusion models to generate 3D Gaussian models. In particular, GaussianDreamer [37] utilizes a 3D diffusion model to generate point clouds for Gaussian initialization and a 2D diffusion model to guide the optimization of 3D Gaussian by providing rich information of geometry and appearance. [17, 34] have exploited the possibility of introducing 3DGS in dynamic scenes. [17] attempts to solve the novel view rendering problem in dynamic scenes by allowing 3D Gaussians to move and rotate over time while maintaining color, opacity, and size. [34] introduces a Gaussian deformation field network to estimate the deformation of 3D Gaussian over time.

For sparse input views problems, FSGS [41] performs a Gaussian unpooling operation during optimization, which explicitly creates Gaussians in 3DGS. Additionally, this method calculates the Pearson correlation, between the target depth estimated by a pre-trained Dense Prediction Transformer (DPT) [1, 23, 24] and the depth of the rendered image, as a regularization term of loss function. SparseGS [35] explicitly prunes floating Gaussians from the model to reduce the corruption caused by floaters and uses the same depth regularization term calculated by Pearson correlation. [7] use $L_1$ distance to formulate the depth regularization term. Our method omits the depth regularization term in the training loss and conducts semantic-label training with side views obtained with our proposed

view sampling method. Furthermore, we utilize depth prior by lifting 2D depth estimation to 3D space and sample a high-quality point cloud from the lift points for 3D Gaussian initialization.

## 3 METHOD

### 3.1 Preliminary

*3.1.1 3D Gaussian Splatting.* 3D Gaussian Splatting (3DGS) [14] is a technique for real-time 3D reconstruction and rendering photorealistic images. Compared with the original NeRF, which usually takes 1-2 days to train a scene and 30 seconds to render an image [12], 3DGS achieves state-of-the-art visual quality while allowing high-quality real-time rendering. In 3DGS, each scene is represented by a large number of 3D Gaussians and each Gaussian is a 3D object in an ellipsoid shape. The 3D Gaussian function, Equation (1), represents the percentage of the opacity of a point in a particular position $x$.

$$G(x) = e^{-\frac{1}{2}(x-\mu)^T \Sigma^{-1}(x-\mu)} \tag{1}$$

$\Sigma$ is a positive semi-definite matrix, representing the covariance of the 3D Gaussian and $\mu$ is the origin of the 3D Gaussian [5]. $\Sigma$ can be decomposed by $\Sigma = RSS^T R^T$, where $R$ is a rotation matrix and $S$ is a scaling matrix. 3D Gaussians will be projected into 2D image space before rendering and 2D covariance matrix $\Sigma'$ is computed as follows:

$$\Sigma' = JW\Sigma W^T J^T, \tag{2}$$

where $J$ is the Jacobian of the affine approximation of the projective transformation, and $W$ is the view transformation matrix [42]. To render a pixel, 3DGS conducts point-based $\alpha$ blending, as follows:

$$c = \sum_{i=1}^{n} c_i \alpha_i \prod_{j=1}^{i-1} (1 - \alpha_j), \tag{3}$$

where $n$ is the number of points, $c_i$ is the color of the $i$-th point, and $\alpha_i$ is given by evaluating a 2D Gaussian with covariance $\Sigma'$ multiplied with a learned per-point opacity.

*3.1.2 Camera Projection.* Camera projection is the process of projecting 3D points into 2D image space by using the camera parameters, including extrinsic and intrinsic. The whole process can be formulated with the following equation

$$V_{world} \cdot M_{ext} \cdot M_{int} = V_{pixel} \tag{4}$$

where $M_{ext}$ is the extrinsic matrix, which represents the positional information of the camera, and $M_{int}$ is the intrinsic matrix, which represents the camera configuration including the focal length and image resolution, etc. $V_{world}$ refers to points in 3D world space, and $V_{pixel}$ refers to points in 2D image space.

### 3.2 Overview

An overview of our method is illustrated in Fig 1. The first part of our method involves a point cloud densification process, which predicts dense depth maps and lifts the dense maps to the 3D world space, obtaining a point cloud that is denser than the output from COLMAP [27–29]. The second part employs a 3D Gaussian splatting model (3DGS) [14]. We sample 3D points from the dense point cloud, obtained in the previous stage, and initialize 3D Gaussians with the sampled points. 3DGS renders images using its default settings. We not only supervise the training using the training views, which

have a ground-truth RGB image, but also train it with side views, obtained with SAOVS, labeled with semantic prior knowledge.

### 3.3 Point Cloud Densification

3DGS utilizes COLMAP [27–29] to generate sparse point clouds, serving as the initial positions of the 3D Gaussians. However, when the training views of a scene are very sparse, COLMAP cannot generate good-quality point clouds, which will hinder scene optimization. To this end, we propose a Point Cloud Densification (PCD) method to construct high-quality point clouds that are denser than the outputs from COLMAP.

*3.3.1 3D Points Lifting.* We use a pretrained Dense Prediction Transformer (DPT) [1, 23, 24] to predict dense depth maps of the training views. Then, we inversely project the depth maps into the 3D world space. Specifically, for each pixel of a depth map, we construct a ray as follows:

$$\mathbf{r}(t) = \mathbf{o} + t \cdot \mathbf{d}, \tag{5}$$

where $\mathbf{o}$ is the origin of the ray, $\mathbf{d}$ is the direction of the ray, and $t$ is the parameter representing the depth. $t$ can be obtained from the depth map and $\mathbf{r}(t)$ represents the position of a point in the 3D world space.

*3.3.2 Point Set Alignment.* Depth maps predicted by monocular depth estimation methods [1, 23, 24] usually provide relative depth information, rather than absolute depth value, causing a scale ambiguity issue [41]. To address this issue, we conduct 3D point set alignment. Specifically, we first lift points that correspond to the SfM points and remove points that are out of the range, between the largest and smallest depth values provided by the SfM method. Then, we estimate a 3D affine transformation between the lifted points and the SfM points. After that, the estimated 3D transformation is applied to all points lifted from the depth maps.

*3.3.3 Point Sampling.* To alleviate the negative effect caused by inaccurate depth estimation, we randomly sample a portion of points from the dense point cloud. The sampled points, as well as the points from the original SfM point cloud, will be used for Gaussian initialization.

Visual results from PCD are demonstrated in Fig. 2. SfM points with 41 input views can be regarded as a high-quality reference, which can improve the performance of 3DGS by up to 4dB [41]. Compared with SfM points from 4 input views, the output from the PCD module provides useful points that are consistent with the original scene. In addition, PCD can generate points in regions where the SfM method cannot detect salient points, facilitating Gaussian initialization in these regions.

### 3.4 Semantic Prior Supervision

Following DietNeRF [13] and SinNeRF [36], we utilize the unseen views, i.e. side views of the training views, to facilitate the optimization of the 3D Gaussian model. SinNeRF aims at solving the single image 3D reconstruction task and the corresponding side view sampling method of SinNeRF, performing random rotation with a fixed camera position as shown in Fig. 3 (b), can only cover a small range of area. DietNeRF targets simple scenes with small objects, whose embedding can be obtained before training. However,

**Inputs**

**3D Gaussians**

Figure 1: Pipeline of our method. We conduct depth estimation on the training-view images and lift the 2D depth maps to a dense point cloud. Points, sampled from the dense point cloud, as well as the point cloud obtained with the SfM method, are used for Gaussian initialization. During optimization, we conduct side view sampling and render side views for regularization.

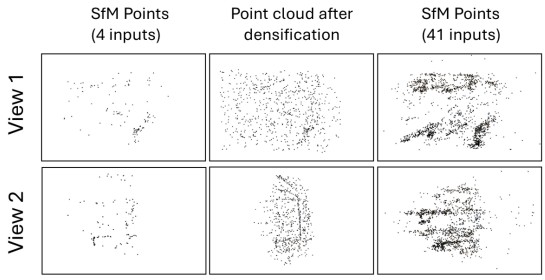

Figure 2: Comparison between the sparse point clouds output from the SfM method, with 4 input views and 41 input views, and our point cloud densification method. As the number of input views increases, the SfM method is capable of producing point clouds of high quality. Compared with the SfM method, PCD can generate accurate points in regions where the SfM method fails, facilitating Gaussian initialization in these regions.

real-life scenes, e.g. scenes in the ScanNet dataset, usually contain different objects in different locations. Therefore, we proposed Systematically Angle of View Sampling (SAOVS), which adaptively samples side views that cover a large portion of the scenes and have a similar angle of view as the training views. An illustration of SAOVS is shown in Fig. 3 (c). SAOVS can be partitioned into two steps, i.e. camera position sampling and camera direction sampling.

*3.4.1 Camera Position Sampling.* In this step, we adopt linear interpolation method to parameterize the position of side-view cameras from Camera $i$ to Camera $i+1$. Given that $t$ is a random variable, the position of the sampled position is represented by a linear function, as follows:

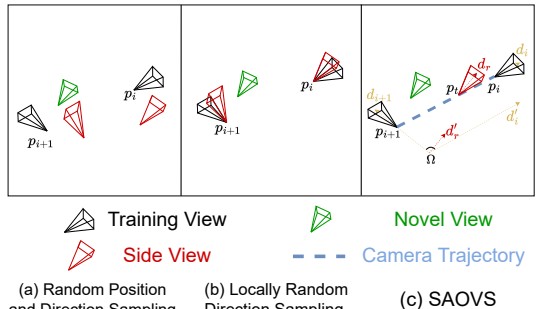

Figure 3: Comparison of different sampling methods. (a) Random sampling of the position and direction of a camera. (b) SinNeRF [36], which fixes the position of the camera and conducts random sampling for the direction. (c) Our sampling method (SAOVS), which allows the direction of the side view to be similar to the training view, compared to (a), and enables the side views to cover more area, compared to (b). $d_i$ and $d_{i+1}$ are the directional vectors of two nearby Cameras $i$ and $i+1$, respectively. $p_i$ and $p_{i+1}$ are the corresponding camera positions. $d_i$, $d_i'$, $d_r$ and $d_r'$ are vectors, such that $d_i \cdot d_i' = 0$ and $d_r \cdot d_r' = 0$. We determine $d_r'$ by using SLERP with the random variable $r$, and $p_t$ by using linear interpolation with the random variable $t$.

$$p(p_i, p_{i+1}; t) = tp_i + (1-t)p_{i+1}, \qquad (6)$$

where $p_i$ and $p_{i+1}$ are the positions of Camera $i$ and Camera $i+1$, respectively.

*3.4.2 Camera Direction Sampling.* We adopt spherical linear interpolation (SLERP) [30] to parameterize the direction of side views.

SLERP is based on the fact that any point on an arc is a linear combination of the two ends, $d_0$ and $d_1$. Given $r$ is a random variable, the direction of the sampled position can be represented as follows:

$$\mathrm{d}(d_i, d_{i+1}; r) = \frac{\sin[r\Omega]}{\sin \Omega} d_i + \frac{\sin[(1-r)\Omega]}{\sin \Omega} d_{i+1}, \qquad (7)$$

where $\Omega$ is the angle subtended by the arc, such that $\cos\Omega = d_0 \cdot d_1$. The concept of SLERP is illustrated in Fig. 3 (c). By using SLERP and controlling the parameter $r$, we can guarantee that the sampled side views have similar contents as the corresponding training views.

*3.4.3 Progressive Training.* We model parameter $t$ and $r$ as random variables subject to two Normal distributions, $\mathcal{N}(0, \sigma_t)$ and $\mathcal{N}(0, \sigma_r)$, respectively. To stabilize the training, we set $\sigma_t$ and $\sigma_r$ to very small numbers, and increase them during training.

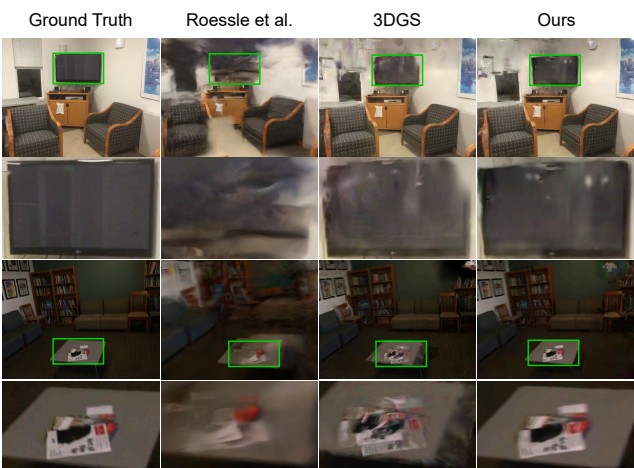

**Figure 4: Novel view synthesis results from different methods, namely Roessle *et al.* [25], 3DGS [14], and our method, on the ScanNet Dataset. With sparse inputs, both [25] and 3DGS struggle to gather enough information to accurately reconstruct challenging areas, such as the television in the first scene. Roessle *et al.* [25], which uses inaccurate dense prediction for regularization, results in ghost effects and blurring in the rendered images. 3DGS does not employ any regularization and consequently fails to capture some details. In contrast, our method successfully renders a sharper image of the television in the first scene and the books in the second scene.**

## 3.5 Training Losses

*3.5.1 Semantic Consistency Loss.* Previous works [13, 36] have shown that the semantic consistency of multi-view data facilitates the reconstruction of the global structure of the scene. We utilize a pretrained DINO-ViT [2] as SinNeRF [36] to extract semantic embedding of cropped regions from a training view and its side view. The semantic consistency loss is defined as follows:

$$\mathcal{L}_{cls} = \|f_{vit}(I_{gt}) - f_{vit}(I_r')\|^2, \qquad (8)$$

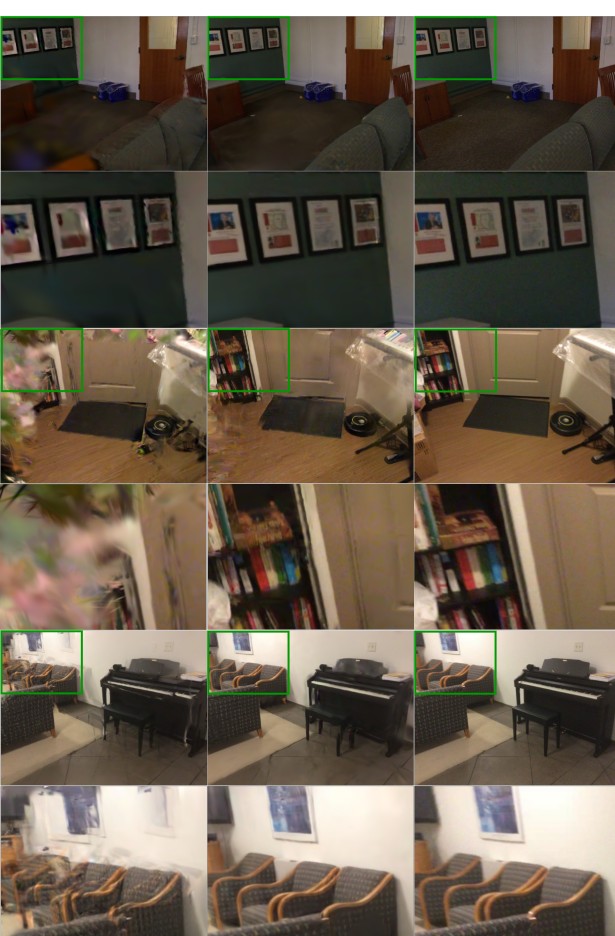

**Figure 5: Novel view synthesis results from our method and 3DGS on the ScanNet Dataset. Compared with 3DGS, our method can generate novel view images with fewer distortions and more photo-realistic details. With semantic consistency loss, our method can reconstruct the geometry of objects, e.g. chairs and couches. With generative adversarial loss, our method can generate fine-grained details, such as frames on walls.**

where $f_{vit}(\cdot)$ refers to the DINO-ViT encoder, $I_{gt}$ is a patch randomly cropped from the training view image and $I_r'$ is the patch rendered from a side view.

*3.5.2 Generative Adversarial Loss.* Generative adversarial networks (GAN) [3, 10, 11, 26], including a generator and a discriminator, are beneficial for reconstructing good-quality images. A well-trained discriminator can distinguish certain patterns on synthetic images, usually caused by corrupted 3D models in radiance field rendering problems. Following SinNeRF, we adopt differentiable augmentation [40], represented by $T$, and formulate the loss function as

**Table 1: Quantitative comparisons between our method, 3DGS and Roessle *et al.* [25] on the ScanNet dataset [8]. The best results are highlighted in red.**

| | PSNR(↑) | | | | | SSIM(↑) | | | | | LPIPS(↓) | | | | |
| --- | --- | --- | --- | --- | --- | --- | --- | --- | --- | --- | --- | --- | --- | --- | --- |
| | 0708 | 0710 | 0758 | 0781 | Avg. | 0708 | 0710 | 0758 | 0781 | Avg. | 0708 | 0710 | 0758 | 0781 | Avg. |
| Roessle *et al.* [25] | 16.17 | 12.64 | 10.03 | 11.07 | 12.48 | 0.58 | 0.45 | 0.41 | 0.43 | 0.47 | 0.65 | 0.76 | 0.78 | 0.78 | 0.74 |
| 3DGS | 19.30 | 17.56 | 18.84 | 19.36 | 18.77 | 0.60 | 0.62 | 0.73 | 0.68 | 0.66 | 0.52 | 0.44 | 0.33 | 0.44 | 0.43 |
| Ours | 24.33 | 18.51 | 19.15 | 19.92 | 20.48 | 0.78 | 0.68 | 0.76 | 0.71 | 0.73 | 0.37 | 0.43 | 0.33 | 0.35 | 0.37 |

**Table 2: Quantitative comparison of different methods on the LLFF dataset [18].**

| | PSNR(↑) | | | | SSIM(↑) | | | | LPIPS(↓) | | | |
| --- | --- | --- | --- | --- | --- | --- | --- | --- | --- | --- | --- | --- |
| | room | trex | horns | Avg. | room | trex | horns | Avg. | room | trex | horns | Avg. |
| NeRF | 16.69 | 13.46 | 13.06 | 14.40 | 0.58 | 0.30 | 0.31 | 0.40 | 0.47 | 0.55 | 0.56 | 0.53 |
| DS-NeRF | 16.79 | 13.87 | 12.93 | 14.53 | 0.56 | 0.31 | 0.28 | 0.38 | 0.52 | 0.67 | 0.64 | 0.61 |
| ViP-NeRF | 22.60 | 19.41 | 19.50 | 20.50 | 0.80 | 0.62 | 0.67 | 0.70 | 0.18 | 0.25 | 0.18 | 0.20 |
| 3DGS | 19.63 | 18.99 | 19.11 | 19.24 | 0.77 | 0.60 | 0.67 | 0.68 | 0.20 | 0.23 | 0.20 | 0.21 |
| Ours w/o PCD | 20.90 | 19.29 | 19.45 | 18.88 | 0.80 | 0.61 | 0.68 | 0.70 | 0.21 | 0.22 | 0.18 | 0.20 |
| Ours w/o SAOVS | 19.61 | 19.79 | 19.78 | 19.73 | 0.78 | 0.63 | 0.71 | 0.71 | 0.18 | 0.19 | 0.19 | 0.19 |
| Ours w/o generative adversarial loss | 20.56 | 19.57 | 19.50 | 18.88 | 0.78 | 0.64 | 0.70 | 0.71 | 0.15 | 0.19 | 0.19 | 0.18 |
| Ours w/o semantic consistency loss | 19.58 | 19.18 | 20.14 | 19.63 | 0.76 | 0.62 | 0.72 | 0.70 | 0.18 | 0.21 | 0.16 | 0.18 |
| Our Full Model | 21.48 | 19.94 | 20.22 | 20.55 | 0.81 | 0.65 | 0.73 | 0.73 | 0.15 | 0.17 | 0.17 | 0.16 |

follows:

$$\mathcal{L}_D = \mathbb{E}_{x \sim p_{data}(x)}[f_D(-D(T(x)))] + \mathbb{E}_{z \sim p(z)}[f_D(D(T(G(z))))], \tag{9}$$

$$\mathcal{L}_G = \mathbb{E}_{z \sim p(z)}[f_G(-D(T(G(z))))], \tag{10}$$

$$\mathcal{L}_{adv} = \mathcal{L}_D + \mathcal{L}_G, \tag{11}$$

where $G(\cdot)$ and $D(\cdot)$ represent the generator, which refers to the rendering process in our algorithm, and the discriminator trained with Hinge loss [16], respectively. $f_D(x) = \max(0, 1 + x)$ and $f_G(x) = x$.

*3.5.3 Total Loss.* The total loss function for the optimization process is expressed as follows:

$$\mathcal{L}_{total} = (1 - \lambda_1)\mathcal{L}_1 + \lambda_1\mathcal{L}_{D-SSIM} + \lambda_2\mathcal{L}_{adv} + \lambda_3\mathcal{L}_{cls}, \tag{12}$$

where $\lambda_1$, $\lambda_2$, and $\lambda_3$ are weighting factors. $\mathcal{L}_1$ is L1 loss and $\mathcal{L}_{D-SSIM}$ is D-SSIM loss used in 3DGS [14].

## 4 EXPERIMENTS

### 4.1 Implementation Details

We test our method on challenging room-scale scenes in the Scan-Net dataset, used by Roessle *et al.* [25], and indoor scenes in the Local Light Field Fusion (LLFF) dataset [19]. For scenes in the Scan-Net dataset, we train the model with about 18 input views. For scenes in the LLFF dataset, we choose only 4 views as input. The total number of training iterations is 30,000. All experiments were conducted on a single NVIDIA RTX 4090 GPU. We initialize both $\sigma_t$ and $\sigma_r$ at 0.09 and increase them every 100 iterations. $\lambda_1$, $\lambda_2$ and $\lambda_3$ are set to 0.2, 1.59 and 0.75, respectively.

### 4.2 Evaluation Protocol

We evaluated our method against other state-of-the-art methods in novel view synthesis tasks. For quantitative comparison, we compute the peak signal-to-noise ratio (PSNR), Structural Similarity

Index Measure (SSIM) [33], and Learned Perceptual Image Patch Similarity (LPIPS) [39] on the RGB images of test views.

### 4.3 Novel View Synthesis Results

Table 1 shows the quantitative comparison between different methods on scenes of the ScanNet dataset. Our method outperforms the other 3DGS methods [14, 25], in all evaluation metrics. Compared with the state-of-the-art method, Roessle *et al.* [25], our method drastically reduces training time from an average of 11 hours to 40 minutes.

Fig. 4 demonstrates the visual results from our method, 3DGS, and Roessle *et al.* [25] in very challenging settings, which contain very sparse input views. Our method synthesizes images with fewer artifacts and retains rich detailed textures. In particular, our method generates a sharper appearance of the television in the first scene and the books in the second scene. We demonstrate more visual results in Fig. 5, which shows that our method can synthesize novel view images with better quality, compared to the baseline method, 3DGS. Particularly, our method can reconstruct difficult regions, with limited scene coverage of cameras.

Table 2 shows the quantitative comparisons on scenes of the LLFF dataset. Generally, our method outperforms the state-of-the-art methods on the LLFF dataset. An exception is ViP-NeRF [31], which performs well in scenes where objects are easily distinguishable from the background, due to the visibility prior. Therefore, ViP-NeRF outperforms our method in room scene by 1.18dB, in terms of PSNR. However, our method synthesizes images with good quality and textures. Our method outperforms 3DGS by 1.31dB, in terms of PSNR, and 0.05, in terms of SSIM.

### 4.4 Ablation Studies

To verify the effectiveness of our proposed components, we conduct ablation experiments on the LLFF dataset. Results in Table 2 and Fig. 7 show that with all components proposed by our method,

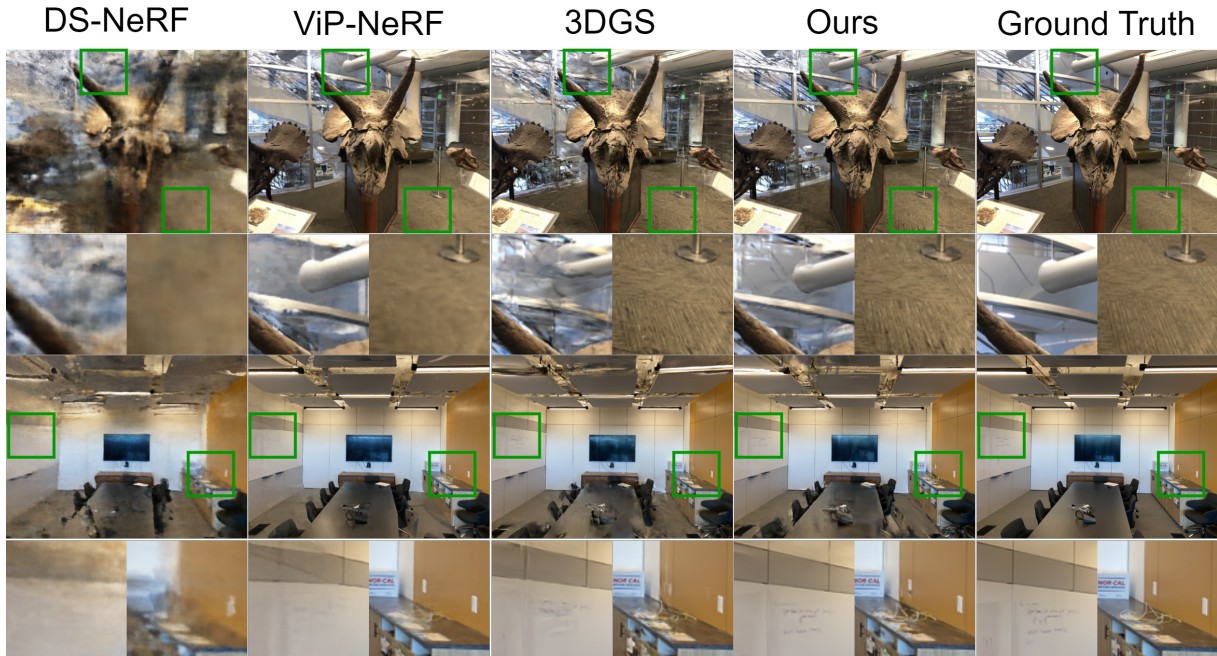

**Figure 6: Novel view synthesis results from different methods, namely DS-NeRF [9], ViP-NeRF [31], 3DGS [14] and ours, on the LLFF Dataset.**

the model achieves the best performance and synthesizes photo-realistic novel view images with fine-grained details.

*4.4.1 Without Point Cloud Densification.* Without the PCD process, 3D Gaussians will be initialized with a sparse point cloud obtained with SfM. Results, shown in Table 2, illustrate that a slight performance drop is recorded if PCD is not performed. Also, we conduct another study to validate the effectiveness of the PCD and investigate the impact of the point cloud sampling rate on the result. Fig. 8 shows that the quality of the point cloud can significantly affect the quality of synthesized images. When sampling points with a sampling rate of 0.1%, the model achieves the best performance. As the sampling rate increases, incorrect depth information generally corrupts the scenes and degrades the performance. Our PCD module can improve the overall performance of the method from the baseline, i.e. without point cloud densification.

*4.4.2 Without SAOVS.* We compare our side-view sampling method with the method shown in Fig. 3 (a), whose direction and position of the side-view cameras are chosen randomly. If the side-view directions are not similar to the training directions, or if the positions of side-view cameras and the corresponding training cameras are not close, the contents in the corresponding rendered images are different. Consequently, the pseudo-label regularization will negatively affect the optimization process. Results in Table 2 show that the performance of the model, randomly sampling side views, has a significant performance drop from the full model. Fig. 7 shows that, compared with the model without semantic losses, the model without SAOVS synthesized images with more severe distortions.

This observation suggests that without an appropriate side-view sampling scheme, semantic losses can negatively impact the synthesized results.

*4.4.3 Without Semantic Consistency Loss.* Without semantic consistency loss, quantitative results in Table 2 recorded a significant performance drop. Visual results in Fig. 7 show that objects, occluded in some views, cannot be well reconstructed because the model does not utilize the semantic consistency loss to regularize the training and fails to restore the geometry of the 3D objects.

*4.4.4 Without Generative Adversarial Loss.* Generative adversarial loss encourages the model to optimize 3D scenes by adding constraints on the details seen from side views. With generative adversarial loss, rendered side-view images tend to have similar detailed textures as training views, encouraging the restoration of fine-grained 3D details. Otherwise, artifacts, corrupting the local textures can be observed from the synthesized visual results as shown in Fig. 7.

## 5 CONCLUSION

In this work, we propose an effective 3D reconstruction method for novel-view synthesis from sparse input views. The proposed method includes a Point Cloud Densification (PCD) module and an enhanced training process, supervised by both the training views with ground truth and the pseudo labels generated by semantic knowledge priors. The PCD module utilizes depth prior, provided by a pretrained Dense Prediction Transformer (DPT), to reconstruct high-quality point clouds. These point clouds are then fed to the

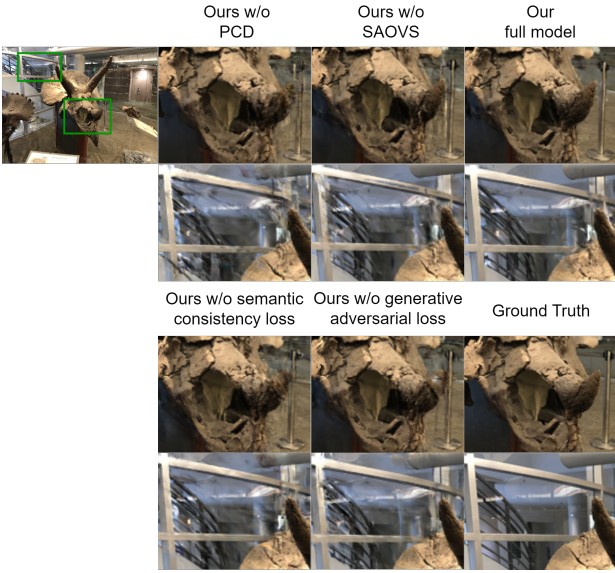

Figure 7: Ablation studies on the LLFF Dataset. In the absence of PCD, we observed a minor decline in performance. Without the semantic consistency loss, objects that are occluded in some views are not effectively reconstructed. If the generative adversarial loss is not applied, artifacts that disrupt the local textures become noticeable in the synthesized visual results. Lastly, without SAOVS, the pseudo-label regularization adversely impacts the optimization process, leading to more pronounced distortions.

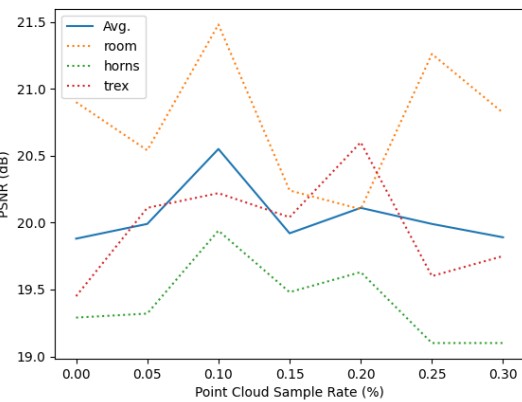

Figure 8: Relationship between the quality of synthesized images and the point cloud sampling rate. Generally, the performance of the model improves as the sampling rate increases. However, when the sampling rate exceeds 0.1%, the performance declines. This is due to the fact that inaccurate depth prediction adversely affects the initialization of 3D Gaussians.

3D Gaussian initialization module, facilitating a better initialization state. In the optimization process, we propose a robust side-view sampling method, called Systematically Angle of View Sampling (SAOVS). This method samples random side views that are similar to the corresponding training views. Specifically, SAOVS combines linear interpolation and spherical linear interpolation (SLERP) to parameterize the position and orientation of the side-view cameras and randomly generate side views. Training with these side views allows the optimized scenes to be semantically consistent from different views and to have fine-grained details. Our experimental results show that our method outperforms the baseline method, 3D Gaussian Splatting, on the ScanNet dataset by 1.71dB and the LLFF dataset by 1.31dB. It also synthesizes photo-realistic novel view results with superior visual quality.

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
