# OpenReview forum: "Point Cloud Densification for 3D Gaussian Splatting from Sparse Input Views"
_acmmm.org/ACMMM/2024/Conference — MM2024 Poster_

### Official Review · Reviewer_JYPy · 2024-05-21

**Rating:** 2
**Confidence:** 4

**Summary:**

A 3DGS-based novel-view rendering method for the sparse-view case is proposed. This method obtains a denser point cloud with the help of the monocular depth estimation method and generates some unseen perspectives other than the train views to assist the training, helping the 3DGS achieve better rendering quality under the sparse view condition.

**Strengths:**

Under sparse-view conditions, the method gets better rendering quality compared to the vanilla 3DGS.

**Limitations:**

The authors claim in the introduction that other methods for sparse-view rely on the prior knowledge provided by the monocular depth estimation and are therefore sensitive to an incorrect depth prior. Yet the authors also use the monocular depth estimation in their paper and just add a random drop strategy. What if the correct depth is randomly dropped, and what is left behind is the incorrect depth? Wouldn't that be even more sensitive to an incorrect depth prior?

In addition, the authors have cited some 3DGS-based works that also deal with sparse-view conditions and have pointed out their problems. However, the authors do not compare the proposed method with these methods in the experiment. Thus, what is said in the introduction lacks experimental support, and there is no evidence showing that the proposed method has solved these problems.
As far as I know, “Depth-Regularized Optimization for 3D Gaussian Splatting in Few-Shot Images” has been accepted by CVPRW and open-sourced before the ACM MM submission deadline, and you don't compare any of them.

In the ablation study, SAOVS is not compared to the methods used in DietNeRF and SinNeRF. Thus, this strategy is not proven to be more effective.

Missing period in line 125.

Why 4 views picked for LLFF and for ScanNet 18 views? Why not conduct more sparse-view case experiments? Besides, to my knowledge, methods based on single-view depth estimation lack multi-view consistency, which may cause degradation when more views are involved. So, does this paper solve this problem?

**Suitability:**

3

---

### Official Review · Reviewer_JHvX · 2024-05-23

**Rating:** 4
**Confidence:** 3

**Summary:**

The paper aims to enhance the performance of 3D Gaussian splatting when working with sparse image inputs.
This is composed of two main parts: densification and additional losses from side view sampling.
It proposes a three-step approach for densifying initial Gaussians: extracting depth maps using a Dense Prediction Transformer (DPF), processing these depth maps to generate additional Gaussians and aligning them using depth values from SfM, and finally, randomly sampling a portion of the generated Gaussians.
To address the limitations posed by sparse input images, the paper also introduces an augmentation policy named Systematically Angle of View Sampling (SAOVS), which samples intermediate views of the training cameras.
Using these side views, the method optimizes Gaussians with two additional losses: consistency loss and adversarial loss.

**Strengths:**

Although the paper demonstrated significant performance improvement, the novelty of each method does not seem significant. The techniques used, while effective, do not introduce substantial new concepts to the field.

**Limitations:**

### Novelty

Although the paper demonstrated significant performance improvement, the novelty of each method does not seem significant. The techniques used, while effective, do not introduce substantial new concepts to the field.

### Clarity of Motivation

The motivation in the introduction could be clearer. Based on the content, the central theme of the paper is overcoming the challenge of neural rendering from sparse input views. The narrative in the introduction should focus on this problem.
Currently, nearly half of the paragraphs in the introduction are heavily focused on "depth." Depth, while important, is not the central theme; it is one of three components (densification, side view sampling, and exploiting depth information).
Emphasizing the challenge of neural rendering from sparse input views and positioning the discussion on "depth" within this context would strengthen the importance of the work. To enhance clarity, the introduction could be more organically structured to emphasize neural rendering from sparse input views, aligning more closely with the main aim of the paper. This would help in making the narrative more focused and cohesive.

**Suitability:**

2

---

### Official Review · Reviewer_BtKo · 2024-05-24

**Rating:** 4
**Confidence:** 2

**Summary:**

The performance of 3D Gaussian Splatting (3DGS) for novel view synthesis degrades when the camera coverage becomes sparse. To address this issue, the paper 1) proposed to improve the input quality for 3DGS with point cloud densification and 2) designed a new side-view sampling scheme, namely Systematically Angle of View Sampling, for regularization. As a result, the proposed method outperformed the compared methods on ScanNet and LLFF datasets.  The effectiveness of 1) and 2) is validated by ablation studies.

**Strengths:**

* The paper proposes a simple yet effective way to address the novel view synthesis problem with sparse views.
* The proposed method has achieved competitive performance against recent work.
* The motivation and design of the method are clear, and its effectiveness is clearly demonstrated.
* The paper is well structured and easy to follow.

**Limitations:**

* There seems to be a lot of literature missing in Section 2.1. Please see the reference list below for some examples.
* More comparisons can be added to Tables 1 and 2. For example, SparseNeRF [6] and FreeNeRF [8] can be added to Table 2 (LLFF dataset) and DaRF [4] and SCADE [5] can be added to Table 1 (ScanNet dataset).
* While the technical contributions of the paper include Point Cloud Densification and Systematic Angle of View Sampling, the title only includes the former.

[1] Mvsnerf: Fast generalizable radiance field reconstruction from multi-view stereo

[2] Nerdi: Single-view nerf synthesis with language-guided diffusion as general image priors

[2] Consistentnerf: Enhancing neural radiance fields with 3d consistency for sparse view synthesis.

[3] InfoNeRF: Ray Entropy Minimization for Few-Shot Neural Volume Rendering

[4] Darf: Boosting radiance fields from sparse inputs with monocular depth adaptation

[5] Scade: Nerfs from space carving with ambiguity-aware depth estimates.

[6] Sparsenerf: Distilling depth ranking for few-shot novel view synthesis.

[7] Diffusionerf: Regularizing neural radiance fields with denoising diffusion models

[8] Freenerf: Improving few-shot neural rendering with free frequency regularization

**Suitability:**

2

---

### Official Review · Reviewer_kZgi · 2024-05-25

**Rating:** 4
**Confidence:** 3

**Summary:**

With sparse input views, the paper presents a 3D Gaussian splatting method for novel view synthesis. The results enhance the PSNR by about 1-2dB compare to the original 3DGS. The paper proposes point cloud densification, systematically angle of view sampling and  generative adversarial loss to improve the accuracy.

**Strengths:**

The paper improves the novel view synthesis with sparse input views than the original 3DGS.

The paper proposes several strategies to improve the rendering quality, such as point cloud densification, systematically angle of view sampling and generative adversarial loss etc.

**Limitations:**

To compare running time with NeRF based methods is not fair, as 3DGS is much more efficient than NeRF. Thus, this is not a merit of the method.

The method relies on pretrained Dense Prediction Transformer to dense the depth map, the generalization ability of the method depends on the pre-trained model. Excepts for the room scenes in the experiments, more different scenes, such as outdoor scenes, man-made objects etc., should also be evaluated. The generalization ability is not clear now.

**Suitability:**

2

---

### Meta-Review · Area_Chair_fkRx · 2024-07-08

**Recommendation:** Accept (Poster)
**Confidence:** 5

**Metareview:**

This paper presents an efficient method to enhance the performance of 3DGS with sparse training views. Instead of applying depth maps for regularization, they proposed a densification method to generates high-quality point clouds for improved initialization of 3D Gaussians. The proposed method is simple but effective. All reviewers have positive final ratings, are satisfied with the response, and recommend accepting the paper. I agree with their recommendation. Thanks for the authors' effort and rebuttal.